# Inflammaging-Driven Osteoporosis: Is a Galectin-Targeted Approach Needed?

**DOI:** 10.3390/ijms26136473

**Published:** 2025-07-04

**Authors:** Marina Russo, Caterina Claudia Lepre, Annalisa Itro, Gabriele Martin, Gianluca Conza, Maria Consiglia Trotta, Monica Puticiu, Anca Hermenean, Francesca Gimigliano, Michele D’Amico, Giuseppe Toro

**Affiliations:** 1PhD Course in National Interest in Public Administration and Innovation for Disability and Social Inclusion, Department of Mental, Physical Health and Preventive Medicine, University of Campania “Luigi Vanvitelli”, 80138 Naples, Italy; marina.russo@unicampania.it; 2School of Pharmacology and Clinical Toxicology, University of Campania “Luigi Vanvitelli”, 80138 Naples, Italy; 3PhD Course in Translational Medicine, University of Campania “Luigi Vanvitelli”, 80138 Naples, Italy; caterinaclaudia.lepre@unicampania.it; 4Department of Experimental Medicine, University of Campania “Luigi Vanvitelli”, 80138 Naples, Italy; michele.damico@unicampania.it; 5Multidisciplinary Department of Medical, Surgical and Dental Sciences, University of Campania “Luigi Vanvitelli”, 80138 Naples, Italy; annalisa.itro@studenti.unicampania.it (A.I.); gabriele.martin@studenti.unicampania.it (G.M.); gianluca.conza@studenti.unicampania.it (G.C.); francesca.gimigliano@unicampania.it (F.G.); giuseppe.toro@unicampania.it (G.T.); 6Faculty of Medicine, Vasile Goldis Western University of Arad, 310144 Arad, Romania; puticiu.monica@uvvg.ro (M.P.); hermenean.anca@uvvg.ro (A.H.); 7“Aurel Ardelean” Institute of Life Sciences, Vasile Goldis Western University of Arad, 310144 Arad, Romania

**Keywords:** osteoporosis, osteoporotic fractures, Galectins (Gals), Galectin-1 (Gal-1), Galectin-3 (Gal-3)

## Abstract

Osteoporosis (OP) is a chronic disease characterized by reduced bone mass and altered microarchitecture, leading to bone fragility and fractures. Due to its high morbidity, disability, and healthcare costs, identifying new biomarkers and therapeutic strategies is crucial for improving OP diagnosis and prevention. In this context, this narrative review aims to depict the role of carbohydrate-binding proteins Galectins (Gals) in the combined processes of inflammation and aging contributing to bone fragility by exploring their potential as novel therapeutic targets for OP.

## 1. Introduction

Osteoporosis (OP) is a chronic musculoskeletal disease characterized by a reduced bone mass and altered microarchitecture, leading to bone fragility [1,2,3] (Figure 1).

OP is the result of an unbalanced bone metabolism, with a bone resorption by osteoclasts higher than bone formation by osteoblasts. Its etiology can be considered multifactorial since hormonal, nutritional, and lifestyle factors play a crucial role in OP onset and progression, as well as genetic modifications [4]. Variants in genes encoding collagen type I alpha 1 (COL1A1), vitamin D receptor, and estrogen receptor have been reported in osteoporotic population [5]. Additionally, some conditions (hyperthyroidism, hyperparathyroidism, and Cushing’s syndrome) could lead to secondary forms of OP [6]. Similarly, diabetes or some autoimmune rheumatologic diseases, such as rheumatoid arthritis, ankylosing spondylitis, and systemic lupus erythematosus, may further increase the OP risk [7,8]. Nutritional deficiencies also play a pivotal role in OP condition: insufficient calcium and vitamin D intake negatively affect bone mineralization [9], and protein deficiency and an excessive intake of phosphorus and sodium can contribute to OP development [10]. Other well-documented risk factors for OP development are physical inactivity (negatively associated with bone density), smoking (contributing to bone oxidative stress), and excessive alcohol consumption (disrupting hormone balance influencing bone homeostasis) [11]. Furthermore, some medications including long-term use of glucocorticoids, anticonvulsants, and certain antiretroviral drugs contribute to bone demineralization and increase the risk of osteoporotic fractures (OF) [12]. These latter are associated with high morbidity and relevant physical disability, leading to reduced quality of life, and high healthcare costs [13,14]. Worthy of note, OF incidence is expected to constantly increase due to the aging of the general population [15,16,17,18]. Overall, despite the high incidence and costs associated with OP, to date a significant percentage of cases remain undiagnosed, only a quarter of diagnosed cases are treated, and an improved drug adherence is still needed [4,19,20].

### The Influence of Inflammaging in OP

After the age of 30, bone mass appears to slowly and progressively decline [21], with a net loss of bone mass and a decreased bone regeneration, due to fewer active osteoblasts combined with a stable or increased osteoclast activity [22,23,24]. Of course, with the increasing age, complex changes negatively affect bone health, leading to an increased OP and OF risk [25]. Indeed, the population over the age of 65 years must undergo in-depth clinical investigations to avoid the onset of fractures [4,26].

From a molecular point of view, aging leads to a state of chronic low-grade inflammation and cellular senescence, also known as “inflammaging”, which is associated with OP [27,28]. This is underlined by biological and molecular mechanisms different from the ones characterizing post-menopausal syndrome or glucocorticoid use-driven OP (Figure 2).

Processes such as oxidative stress, cytokine storms, DNA damage, autophagy defects, nutrient deprivation/reduced metabolic cell activity, and mitochondrial dysfunction lead to bone inflammaging [29], underlined by the following molecular mechanisms: cell cycle arrest through the upregulation of p16, p21, p53 (markers of DNA damage response); impaired autophagy by the activation of mTOR (an autophagy inhibitor); promotion of senescence by decreased levels of Sirtuins (able to counteract osteoblast senescence); increased type 1 interferon and STING pathway, linking DNA damage, inflammation, and cellular senescence [30].

Aging can also modify hormonal balance, thus contributing to OP [31]. In this regard, the reduction in estrogen in women after menopause and in testosterone in older men affects the activity of osteoclasts and osteoblasts by leading to bone alterations [32,33]. Conversely, the increased levels in the elderly population of parathyroid hormone (PTH), produced by the parathyroid glands, stimulates the activity of osteoclasts, leading to bone weakness [34]. It is well known that aging leads to a reduction in the intestinal absorption of calcium and vitamin D [35], a decreased efficiency in producing vitamin D through sun exposure, and a lower ability of the kidney to convert vitamin D to its active form [36]. Consequently, the body tends to mobilize calcium from the bones to maintain normal blood levels, contributing to bone loss [37]. Another factor contributing to age-related OP development is the progressive loss of muscle mass, known as sarcopenia. Indeed, the muscles play a fundamental role in supporting bones and maintaining their balance and strength, with a bone–muscle molecular crosstalk having a relevant role in determining bone quality [38]. Therefore, the risk of bone fractures increases with muscle mass loss [39].

Overall, the combination of imbalances in bone remodeling, hormonal changes, reduced nutrient absorption, and lifestyle factors in elderly people contributes to progressive bone loss, which can be modified both by lifestyle changes (proper diet, regular physical activity) and pharmacological options.

## 2. OP Treatments: Approved and Ongoing

Regarding the pharmacological options used in the prevention and management of OP and OF, these include antiresorptive drugs, anabolic agents (AAS), and dual-action agents [40,41].

Bisphosphonates are antiresorptive drugs which reduce bone resorption mainly inducing osteoclast apoptosis, and currently represent the standard pharmacological OP therapy [42]. All approved bisphosphonates have been shown to reduce the risk of vertebral fractures and increase bone mineral density [43], with their antiresorptive effect persisting even after their discontinuation [43,44]. Bisphosphonates are taken by oral or intravenous administration. Particularly, alendronate, risendronate, and ibandronate are daily, weekly, or monthly administered, while ibandronate and zoledronate are intravenously given, respectively, every 3 months and once a year [45]. These drugs are characterized by limited toxicity, with the main side effects concerning the upper gastrointestinal tract for oral bisphosphonates, and flu-like symptoms (fever, muscle, and bone) after the first infusion for intravenous bisphosphonates, treated with symptomatic drugs [46,47]. However, bisphosphonates cannot be administered in patients with serious kidney problems [48]. In rare cases and when used for a long time, they have been associated with osteonecrosis of the jaw (caused primarily by avascular necrosis and complicated by secondary infection) and atypical femur fractures [49,50]. Another antiresorptive drug commonly used for the treatment of OP is Denosumab, a monoclonal antibody that binds strongly and specifically to the so-called RANK ligand (RANKL), a key mediator of bone resorption [51,52]. Particularly, Denosumab blocks the interaction between RANKL and RANK on osteoclasts, thus inhibiting their differentiation, activation, and survival [53].

Regarding the AAS, such as the parathyroid hormone analogues teriparatide and abaloparatide, they are used in the OP treatment to stimulate bone formation [54]. Indeed, daily AAS administrations promote osteoblast activity and are generally used in patients with severe OP or not responder to other treatments [55].

More recently, the dual action agent Romosozumab was introduced as a drug able to both stimulate bone formation and reduce bone resorption by acting on the Wnt pathways [56]. This monoclonal antibody, administered as a monthly subcutaneous dose, acts by binding and inhibiting sclerostin (an important regulator of bone formation), leading to a significantly reduction in both vertebral and non-vertebral fractures in recent clinical studies [57,58,59,60,61].

Another pharmacological option in OP management is hormone replacement therapy (HRT). Acting on the hormone imbalance, HRT may consist of taking estrogen alone or in combination with other sex hormones (progestins). This treatment slows bone turnover and increases bone mineral density in all skeletal areas in postmenopausal women of any age [62]. Other OP drugs are selective estrogen receptor modulators (SERMs), binding to estrogen receptors and acting as agonists or antagonists, depending on the organ [63]. Although not a first-line OP treatment, SERMs (tamoxifene, bazedoxifene, and raloxifene) are used in postmenopausal women with both OP and increased risk of breast cancer, decreasing bone resorption and the risk of vertebral fractures [64,65].

Besides the OP drugs currently available, calcium and vitamin D supplementation are certainly useful to reduce OF [66] and should be always recommended [67,68]. Similarly, bone loss and OF-related pain are reduced by calcitonin supply, able to maintain calcium homeostasis, and counterbalance the aged-induced PTH increase [66,69].

Some other therapeutic options are currently under clinical investigation. A randomized control clinical trial evidenced the efficacy of Xulin Jiangu granules, a traditional Chinese medicine, in postmenopausal OP aggravated by renal failure and blood stasis syndrome (NCT03563235). Moreover, the effects of 24-month application of nitroglycerin ointment have been investigated in OP in elderly women (NCT00252421). Great interest has also been recently generated by mesenchymal stem cells (MSCs), since they can be useful in halting OP deterioration by restoring bone turnover, re-establishing the balance between bone formation and resorption, increasing bone mineral density, and reducing bone inflammation [70]. Particularly, a clinical study is testing the intravenous infusion of autologous fucosylated bone MSCs in patients with OP (NCT02566655), while a phase 2 clinical trial, still recruiting patients, aims to evaluate the effectiveness of the allogeneic mesenchymal cell from umbilical cord implanted in OP patients (NCT04501354). Furthermore, the application of a local osteo-enhancement procedure in the proximal femur of OP women is under investigation in a clinical study currently in the recruitment phase (NCT05202678).

From a preclinical point of view, the extracts of Epimedium (Berberidaceae), also known as horny goat weed or Yin Yan Huo, demonstrated several protective effects on musculoskeletal system as a putative modulator of estrogen signaling, RANKL/RANK or reactive oxygen species (ROS) pathways [71,72,73]. In particular, the Epimedium extracts epimedin B and epimedin C significantly stimulated the proliferation of osteoblast-like cells (UMR106) [72,74]. An ovariectomized rat model with OP was treated with icariin, extracted from Epimedium, which inhibited bone loss [75]. Epimedium flavonoids were also used to treat rats in an in vivo model of retinoic acid-induced OP, leading to increased bone mass and improved biomechanical properties of the bone [76].

While bisphosphonates and denosumab are used for OP caused by post-menopausal syndrome, use of glucocorticoids or secondary OP, the strategies aimed to specifically reduce inflammaging-driven OP and improve its outcome are still under investigation. Indeed, only a single recent study associated icariin with an anti-inflammaging effect through the modulation of autophagy in an animal OP model [77].

## 3. Galectin Family and Inflammaging

Galectins (Gals) are a family of carbohydrate-binding proteins that play a crucial role in the regulation of various cellular processes including aging, inflammation, immune modulation, and tissue remodeling [78]. The family includes 15 members (Figure 3) with a broad tissue distribution, located mainly in the cell cytoplasm, but also translocated into the nucleus [79].

Gal functions can vary depending on the cell-type involved, mainly immune and inflammatory [80]. Overall, Gals are able to regulate cell-cell interaction, apoptosis, epithelial mesenchymal transition, tumor progression, and, particularly, inflammation, aging, and age-related diseases [80].

Particularly, during acute inflammation, Galectin 1 (Gal-1) seems to exert an anti-inflammatory role by controlling neutrophil trafficking and extravasation [81], while Galectin 3 (Gal-3) is characterized by pro-inflammatory actions through the increase in macrophage and neutrophil numbers and phagocytosis [82,83].

In the wide area of inflammaging-related chronic disorders, Gal-1 seems to play a controversial role. Indeed, it has been proposed as an attractive immunosuppressive agent, able to reduce T helper 1 (Th1) and 17 Th17 pro-inflammatory responses and shift the cytokine balance toward a T helper 2 (Th2)-dependent anti-inflammatory polarized profile in experimental models of experimental autoimmune encephalomyelitis (EAE) [84], inflammatory bowel disease [85], Graft versus host disease [86], and experimental autoimmune uveitis [87]. On the contrary, a pro-fibrotic role has been associated with Gal-1 increase both in cardiac and liver fibrosis induced by diabetes [88,89].

Galectin 2 (Gal-2) could exert a protective role during allergic inflammation by inducing apoptosis of CD8+ T cells [90], while Gal-3 seems to exacerbate asthma, atopic dermatitis, and EAE by promoting an (Immunoglobulin E) IgE increase [91,92,93]. Moreover, the increase in Gal-3 and Gal-1 in bronchoalveolar lavage from Coronavirus disease 2019 (COVID-19)-infected patients was correlated with pro-inflammatory mediators favoring lung fibrosis [94].

Galectin 4 (Gal-4) is associated with an increase in the pro-inflammatory interleukin 6 (IL-6) during colitis [95], while Galectin 9 (Gal-9) is associated with an increase in the anti-inflammatory role during EAE by favoring the apoptosis of Th1 cells [96].

Considering more specifically the aging mechanisms, Gal-3 has been recently identified as a receptor for advanced glycosylation end-products (AGEs) [97]. Accordingly, serum Gal-3 concentration significantly correlated with frailty in elder people [98], with lower Gal-3 levels associated with successful ageing [99]. An increase in Gal-3 has also been correlated with neuroinflammation and neurodegeneration in an experimental model of aged-induced neurodegeneration [100]. Conversely, Gal-3 genetic deletion seems to exacerbate age-related myocardial hypertrophy and fibrosis in mice [101]. In Alzheimer’s disease (AD), the number of microglial cells expressing Gal-1 and Gal-3 tends to increase with age [102,103]. Moreover, microglia-derived Gal-9 seems to favor amyloid-β accumulation in experimental AD [104] (Figure 3).

### 3.1. Gal-1 and Gal-3 in Bone Fragility, Resorption, and Senescence

Except for a single study reporting an association between Galectin 8 (Gal-8) with hyperactive osteoclast phenotype and increased bone resorption [105], Gal-1 and Gal-3 have emerged as major modulators of bone homeostasis. Indeed, Gal-3 showed the highest specificity for bone-related tissues among all the Galectin family members. Considered a marker of chondrogenic and osteogenic cell lineages, Gal-3 is expressed by chondrocytes, osteoclasts, and osteoblasts [106,107,108,109] and is able to interact with AGEs in osteoblast-like cells [110]. In addition, Gal-1 and Gal-3 are the main Gals expressed by bone marrow mesenchymal stem cells (BMSCs), able to differentiate in osteoblasts and regulate bone formation [111]. Worthy of note, Gal-1 and Gal-3 secretion from BMSCs seem to be involved in regulation of osteogenic differentiation and resolution of inflammation [112,113,114,115]. Compared to other Gals, both Gal-1 and Gal-3 can strictly modulate the T-cell immune response [116,117], whose dysregulation contributes to OP progression [118]. Moreover, Gal-1 and Gal-3 have also been reported to show the highest affinity for human factor VIII (FVIII) [119], closely related with mineralization and bone remodeling [120,121].

To date, the evidence regarding the specific functions of Gal-1 and Gal-3 in bone homeostasis emerged from several preclinical settings is controversial. It is supported by limited clinical studies and necessitates further validation to elucidate the potential impact of Gal-1 and Gal-3 modulation in OP patients (Figure 4, Table 1).

A decline in Gal-1 serum levels in aged mice and humans was associated with age-related trabecular bone loss [122]. A study using Gal-1 knockout (KO) mice evidenced a decreased mineral density and alterations in trabecular microarchitecture in aged animals [112]. Moreover, BMSCs isolated from femur and tibia of Gal-1 KO mice exhibited a reduced differentiation into osteoblasts, partially restored by Gal-1 supplementation [112]. This latter evidence was lately confirmed by Takeuchi et al. (2024), who reported that both human and mouse BMSCs treated with Gal-1 showed a reduced osteoclast formation and bone resorption activity [123], suggesting a positive role for Gal-1 in regulating bone turnover. Furthermore, the equine BMSCs cultured in inflammatory conditions showed reduced Gal-1 levels: this could limit their intra-articular repair properties, due to a reduced differentiation into osteoblasts [114]. On the contrary, a negative role of Gal-1 in bone homeostasis has been suggested by an in vitro study reporting a decreased expression of osteocalcin [124], a key marker of osteoblast maturation [129], in BMSCs treated with Gal-1. This limited matrix mineralization.

Concerning Gal-3, the literature mainly describes a negative role in bone homeostasis (Table 1). Indeed, in response to high AGEs levels, Gal-3 levels increased in osteoblast-like cells [110] and inhibited human osteoblast differentiation by modulating the Notch signaling pathway [125]. Gal-3 also promoted osteoclast differentiation, thus influencing the extent of bone resorption [125]. In line with these data, Gal-3 inhibition in human fetal osteoblast cell line hFOB 1.19 promoted the proliferation and differentiation of osteoblasts, improving bone mineralization [126]. Similarly, Gal-3 KO mice exhibited increased osteoblastogenesis, resulting in preserved or increased bone mass [127]. Conversely, a positive role of Gal-3 as a novel regulator of osteoblast–osteoclast interaction has been recently proposed by Simon et al. (2017), who identified the secretion of Gal-3 by osteoblasts as a novel mechanism to control osteoclastogenesis and to maintain trabecular bone homeostasis [128].

### 3.2. Contribution of Gal-1 and Gal-3 to Secondary OP

Gal-1 and Gal-3 implication has been also underlined in several inflammaging-related pathologies contributing to OP, such as diabetes, obesity, and rheumatoid arthritis. In this regard, Gal-3 seems to induce inflammation and death of β-cells in pancreatic islets in patients with type 2 diabetes mellitus [130], with a specific role in the progression of prediabetes to diabetes stage [130] and diabetic nephropathy [131]. Conversely, Gal-1 is upregulated in the early stages of diabetic retinopathy (DR) and in its progression, while it is downregulated in the ocular microenvironment of non-retinopathic diabetic patients [79]. Furthermore, recent studies have highlighted the involvement of Gal-1 in liver and cardiac fibrosis induced by chronic diabetes [88,89]. Elevated Gal-1 and Gal-3 levels have been also reported in obese subjects [132]. Gals are also involved in arthritis, an inflammatory joint disease classified in rheumatoid arthritis (RA) and osteoarthritis (OA) [133,134,135]. Gal-1, often upregulated in RA, has been shown to positively correlate with markers of inflammation such as erythrocyte sedimentation rate and disease activity scores [136]. Its elevated levels may contribute to the regulation of immune responses in RA, potentially exerting anti-inflammatory effects in response to certain treatments [137].

Gal-3 levels are higher in RA than in OA, with Gal-3 substantially expressed and released from the inflamed synovial membrane in RA patients. These can trigger the release of pro-inflammatory cytokines and chemokines from rheumatoid fibrocyte-like synoviocytes [137,138,139]. Furthermore, Gal-3 plays a role in tissue damage through promoting the production of matrix metalloproteinases (MMPs), which contributes to joint degradation.

All over, these observations could lead to considering both Gal-1 and Gal-3 among the factors associated with the inflammaging alterations. Therefore, the ability to selectively modulate Gal-1 and Gal-3 may pave the way for a new therapeutic tool for bone health.

## 4. Perspective of Gal-1/Gal-3 Modulation in OP

To date, no specific Gal-1 and Gal-3 modulators have been tested in OP. Indeed, in OP animal models, the inhibition of Gal-1 and Gal-3 has been obtained by gene silencing [112,127,128]. However, several Gal inhibitors have been developed, such as tricyclic carbohydrate–benzene hybrids, natural carbohydrate-derived ligands, iron-containing glycomimetics based on lactose scaffolds, and thiodigalactosides (TDG) [140], or identified. Interestingly, these compounds are able to target several processes strictly related to inflammation and senescence. This suggests a potential benefit for their specific application in OP driven by inflammaging, in relation to the nonspecific use of bisphosphonates and denosumab in OP caused by post-menopausal syndrome, use of glucocorticoids, or secondary OP.

To this regard, Gal-3 expression seems to be reduced in preclinical models of atherosclerosis by melatonin treatment, leading to decreased inflammation and improved autophagy in macrophages [141]. In cardiac settings, Gal-3 inhibition has been obtained by the increase in the antioxidant protein peroxiredoxin-4 [142] and by the modified citrus pectin [143], with the consequent reduction in oxidative stress and inflammation. Of interest, the Gal-3 inhibitor G3P-01 (a pectin present in fruits and vegetables) has been orally administered for 4 months in aged men and women with elevated serum Gal-3 levels, as a nutritional dietary supplement aimed to favor a healthy aging process (NCT06398821). Therefore, from these preclinical and clinical studies, Gal-3 inhibition could emerge as a useful strategy aimed to specifically target several contributors to inflammaging-driven OP such as the low-grade inflammatory state, the oxidative stress, the ageing processes, and also the deleterious effects of reduced authophagy [144,145]. Indeed, an upregulated autophagic process favors the homeostasis of osteoblasts by promoting their survival (through reduced oxidative stress levels) and mineralization [146,147], overall improving trabecular bone mass and bone formation [148,149,150,151].

Regarding the other Gal-inhibitors under clinical investigation, the selective Gal-3 inhibitors GR-MD-02 (8 mg/kg) (NCT03809052, NCT04607655) or GB1211 (10 and 100 mg/kg) were administered orally, twice daily for 12 weeks, to patients with non-alcoholic steatohepatitis (NASH) (NCT02421094). A phase 2 clinical study is also ongoing in patients with idiopathic pulmonary fibrosis in order to evaluate the efficacy, pharmacokinetics, and pharmacodynamics of another Gal-3 inhibitor, GB0139, previously known as TD139, inhaled in the form of dry powder [152]. Worthy of note, a potential link between deteriorated bone health and fibrosis has been recently highlighted. Indeed, pulmonary fibrosis has been proposed as a risk factor for OP incidence in senile patients, showing a markedly reduced bone mineral density [153]. An attenuated bone architecture and a decreased bone density have been found also in aged patients affected by liver fibrosis [154,155]. Moreover, since fibrosis is a process strictly interrelated with senescence and inflammation [156,157,158], the results of the clinical studies assessing the efficacy and safety profile of Gal-3 inhibitors in fibrotic diseases could be useful also if translated to OP induced by inflammaging. Of interest, a specific anti-fibrotic effect has been shown also by the selective Gal-1 inhibitor OTX008 (or calixarene 0118) in human retinal pigment epithelium cells (ARPE-19) exposed to high glucose at different time-points, as a model of DR. The inhibition of Gal-1 by OTX008 preserved the integrity and functionality of retinal cells by reducing diabetes-induced fibrotic process [159]. A similar effect was obtained in an animal model of cardiac and liver fibrosis induced by chronic diabetes [88,89]. Although no clinical studies aimed to assess the anti-fibrotic effects of OTX008 has been conducted, a Phase I clinical study has been reported to test OTX008 subcutaneously administered (65 mg/day, daily, for 3 weeks) in patients with advanced solid tumors (NCT01724320). This starting for the preclinical evidence shows a reduction in tumor angiogenesis exerted by OTX008 through the inhibition of endothelial cell migration and the reduction in antitumor immune responses by the modulation of the tumor microenvironment [160,161]. Since elevated levels of both Gal-1 and Gal-3 are exhibited by patients with osteosarcoma [162,163,164,165], and long-term survivors of osteosarcoma show higher risks for OP prevalence and OF [166,167], a multifactorial approach aimed to target these Gals in osteoporotic patients with osteosarcoma, whose progression is favored by a state of chronic inflammation and aging/senescence-induced genes [168,169], could be hypothesized as an innovative therapeutic approach.

However, the clinical application of Gal-1 and Gal-3 inhibitors in inflammaging-related OP could be affected by some limitations, such as limited oral bioavailability, poor pharmacokinetic profiles, limited selectivity, challenges in assessing their affinity, and also in reaching specific target cells [140,170].

From the Gal-1 and Gal-3 activation point of view, their expressions have been increased only by specific gene promoters in cancer settings [171,172,173] and, to our knowledge, no specific activators have been tested either in OP or different diseases.

## 5. Conclusions

To date, there are contradictory preclinical data regarding the role of Gal-1 and Gal-3 in bone homeostasis, along with the limited availability of clinical studies and the absence of Gal specific modulator for bone tissue.

However, due to the several pieces of evidence highlighting the role of Gal-1 and Gal-3 in the modulation of inflammation, aging, and senescence, the need to approach Gal-1 and Gal-3 as novel targets of inflammaging-driven OP seems to be necessary. This could be achieved by improving the oral availability of the Gal-inhibitors already developed; by identifying the best candidate in terms of affinity, selectivity, and toxicity in OP preclinical settings; and by translating its potential application in OP patients by evaluating its pharmacokinetic profile, safety, and efficacy.

## Figures and Tables

**Figure 1 ijms-26-06473-f001:**
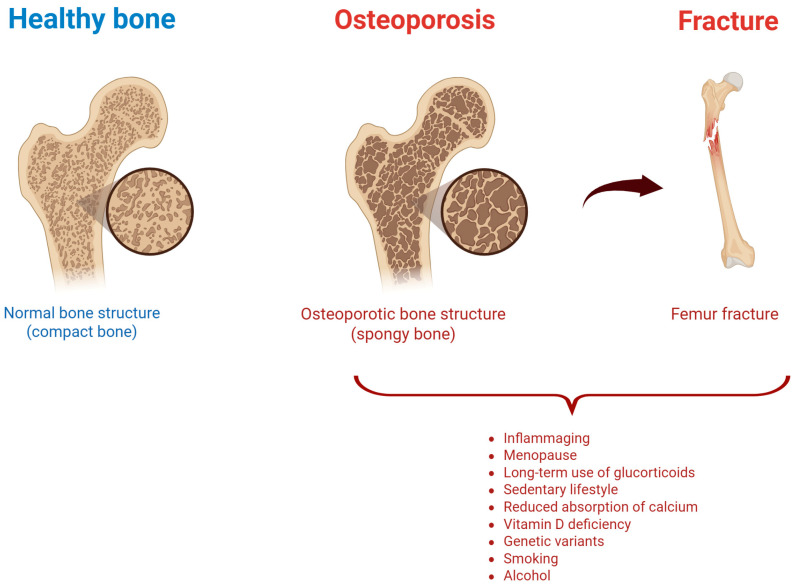
Healthy vs. osteoporotic bone, and risk factors associated with osteoporotic fracture. Created by Biorender.

**Figure 2 ijms-26-06473-f002:**
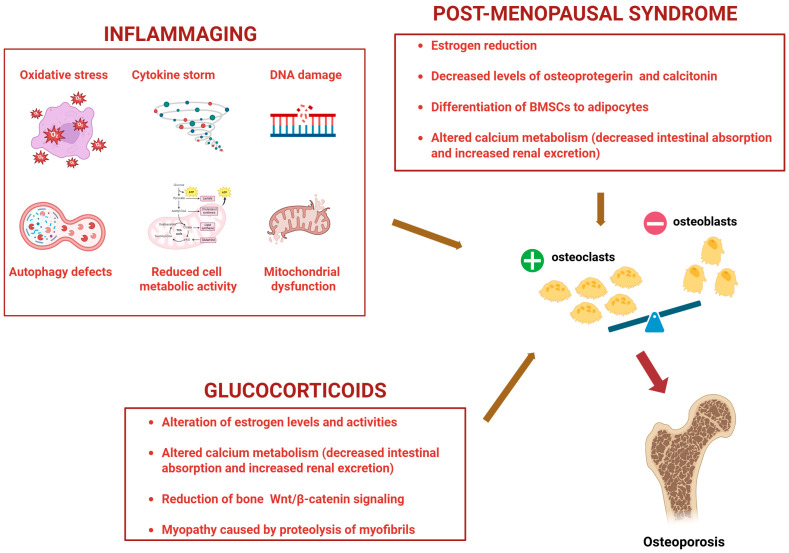
Molecular mechanisms underlying Osteoporosis driven by inflammaging, post-menopausal syndrome, and use of glucocorticoids. BMSCs: Bone marrow mesenchymal stem cells. +: increased; −: decreased. Created by Biorender.

**Figure 3 ijms-26-06473-f003:**
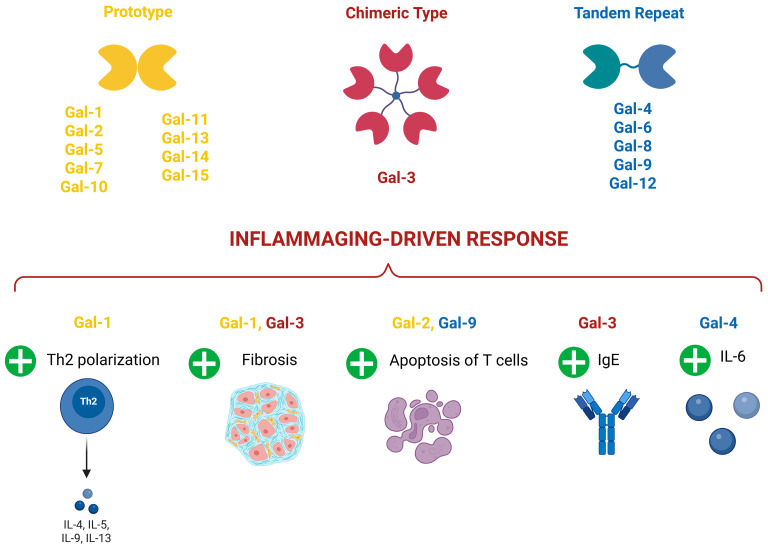
Galectin types and their involvement in inflammaging-driven response. Gal-1: Galectin 1; Gal-2: Galectin 2; Gal-3: Galectin 3; Gal-4: Galectin 4; Gal-5: Galectin-5; Gal-6: Galectin 6; Gal-7: Galectin 7; Gal-8: Galectin 8; Gal-9: Galectin-9; Gal-10: Galectin 10; Gal-11: Galectin 11; Gal-12: Galectin 12; Gal-13: Galectin 13; Gal-14: Galectin 14; Gal-15: Galectin 15; Th2: T helper 2 lymphocytes; IgE: Immunoglobulin E; IL-4: Interleukin 4; IL-5: Interleukin 5; IL-6: Interleukin 6; IL-9: Interleukin 9; IL-13: Interleukin 13. +: increased. Created by Biorender.

**Figure 4 ijms-26-06473-f004:**
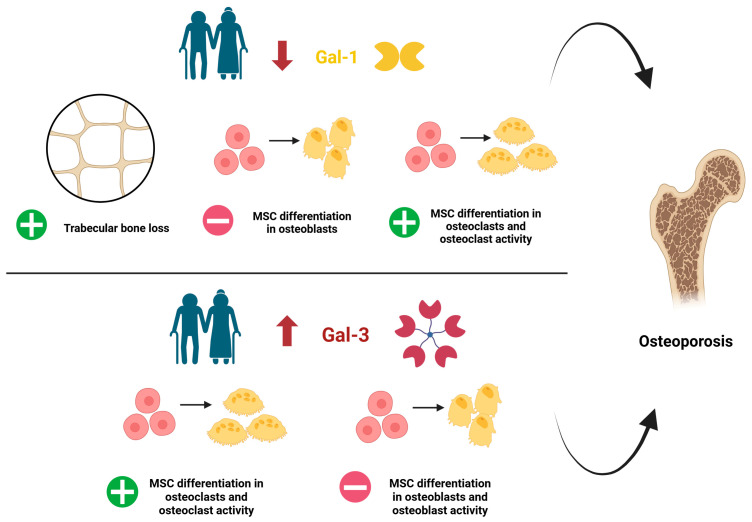
Dysregulation of Galectin 1 (Gal-1) and Galectin 3 (Gal-3) during ageing and main actions precipitating OP. Note that there is controversial evidence regarding the role of Gal-1 and Gal-3 in MSC osteoblast differentiation, as reported in Table 1. MSC: mesenchymal stem cells. +: increased; −: decreased. Created by Biorender.

**Table 1 ijms-26-06473-t001:** Preclinical and clinical studies on Gal-1 and Gal-3 in OP settings.

Reference	Study	Experimental Setting	Treatment	Main Results
Xu et al., 2021 [122]	In vivo Clinical	Aged male Balb/c and C57BL/6 mice Aged osteoporotic patients	/ /	Serum Gal-1 was reduced in aged mice and osteoporotic patients. Gal-1 decline was associated with trabecular bone mass loss in both preclinical and clinical settings
Chen et al., 2022 [112]	In vivo In vitro	Aged and young Gal-1 KO mice BMSCs isolated from femur and tibia of Gal-1 KO mice	/ Gal-1 0.5 μg/mL for 48 h	Deletion of Gal-1 in mice resulted in bone loss, due to a reduced ability of BMSCs to differentiate into osteoblasts. This was more evident in aged mice compared to young ones. In vitro, Gal-1 facilitated the differentiation of BMSCs into osteoblasts
Takeuchi et al., 2024 [123]	In vitro	Osteoclasts differentiated from Human PBMCs Osteoclasts differentiated from Raw 264.7	On both cell types: Recombinant Gal-1 protein, 10 µg/mL for 14 days	Recombinant Gal-1 inhibited osteoclast formation and bone resorption activity
Andersen et al., 2003 [124]	In vitro	Human BMSCs	Gal-1 recombinant (10–1000 ng/mL)	Gal-1 reduced Osteocalcin expression, suggesting a reduction in HBMSC differentiation in osteoblasts
Mercer et al., 2004 [110]	In vitro	MC3T3E1	100–200 microg/mL AGEs-BSA	Intracellular Gal-3 was increased by AGEs
Nakajima et al., 2016 [125]	In vitro	Raw 264.7 Human osteoclast precursors hFOB1.19	Cells exposed to full-length and cleaved Gal-3 secreted from breast and prostate cancer cells	Gal-3, through its interaction with the protein myosin-2A, promoted osteoclast differentiation. Furthermore, the cleaved Gal-3 influenced the extent of bone resorption
Nakajima et al., 2014 [126]	In vitro	hFOB1.19	Recombinant human Gal-3, 1.6 µM, every 3 to 4 days, for 3 weeks Lactose (75 mM), a sugar Gal-3 inhibitor	Gal-3 inhibited osteoblast differentiation through the Notch signaling pathway, and impaired bone formation by reducing the expression of genes implicated in osteoblastic differentiation, such as *RUNX2*, *SP7*, *ALPL*, *COL1A1*. Gal-3 inhibition promoted hFOB1.19 proliferation and differentiation
Maupin et al., 2018 [127]	In vivo	Gal-3 KO mice	/	Gal-3 KO mice exhibited preserved or enhanced bone mass, due to increased osteoblastogenesis.
Simon et al., 2017 [128]	In vivo In vitro	Gal-3 KO mice Osteoclasts and osteoblasts differentiated from Gal-3 KO BMSCs	/ /	Gal-3 KO mice exhibited elevated osteoclast numbers and a lowered trabecular bone volume. Gal-3 secreted by osteoblasts inhibited osteoclast formation

AGEs: Advanced Glycation End products; ALPL: Alkaline phosphatase; BMSCs: Bone Marrow Stromal Cells; BSA: Bovin Serum Albumin; COL1A1: Collagen type I alpha 1; Gal-1: Galectin 1; Gal-3: Galectin 3; hFOB1.19: human fetal osteoblastic cells; KO: knockout; MC3T3E1: mouse calvaria-derived osteoblasts; PBMCs: Peripheral Blood Mononuclear Cells; Raw 264.7: mouse osteoclast precursors; RUNX2: RUNX family transcription factor; SP7: SP7 transcription factor.

## Data Availability

Not applicable.

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
