# Peer review of "Inflammaging-Driven Osteoporosis: Is a Galectin-Targeted Approach Needed?"

_ijms, 2025, doi:10.3390/ijms26136473_

Round 1
Reviewer 1 Report
Comments and Suggestions for Authors
This review elucidates the emerging and conceptually significant roles of Galectin family proteins, particularly Gal-1 and Gal-3, in relation to osteoporosis (OP) associated with inflammaging. The manuscript is generally well-organized and provides comprehensive mechanistic insights, supported by figures and tables. However, several concerns outlined below needed to be addressed to improve clarity and readability.
- The manuscript introduces inflammaging as a significant contributor to age-related osteoporosis, it does not sufficiently differentiate the pathophysiological mechanisms associated with inflammaging-driven osteoporosis from those of other prevalent forms, such as postmenopausal or glucocorticoid-induced osteoporosis. A direct and systematic comparison, preferably illustrated through a summary table or schematic figure, would enhance readers' comprehension of the distinct etiological factors, cellular pathways, and therapeutic implications associated with inflammaging-driven osteoporosis.
- The review offers a comprehensive examination of Gal-1 and Gal-3 within the framework of aging and bone biology. However, it lacks a comparative analysis of Galectin-targeted therapies in relation to established osteoporosis (OP) treatments, such as bisphosphonates and denosumab. To strengthen the central argument, the authors should include a more detailed discussion regarding the potential benefits and limitations of Galectin, particularly in the context of inflammaging. Furthermore, the manuscript should elucidate why Galectins may represent more appropriate therapeutic targets for OP driven by inflammaging, as opposed to general OP, and should be supported by mechanistic or preclinical evidence.
- The review primarily addresses Gal-1 and Gal-3; however, it fails to elucidate the rationale for the selection of these specific subtypes from the broader Galectin family in relation to osteoporosis. A comprehensive discussion regarding the structural features, receptor affinities, tissue-specific expression, or signaling pathways that distinguish Gal-1 and Gal-3 from other Galectins would significantly strengthen the biological justification for their consideration as leading therapeutic candidates.
- The manuscript offers valuable insights into the role of Galectins in OP; however, it does not adequately address the limitations present in the current study. It is recommended that the authors include a concise discussion of the primary knowledge gaps, including the limited availability of clinical studies, the presence of contradictory preclinical data, and the absence of Galectin-specific modulators for bone tissue. Additionally, it is suggested that the authors propose clear future directions to facilitate translational research in this area.
The manuscript is generally well-written and demonstrates a high quality of English; however, it requires proofreading to improve its readability for the readers.
Author Response
Reviewer 1
The Authors thank the Reviewer for appreciating the Manuscript and for his suggestions in order to improve its quality. All the changes to the Manuscript have been highlighted for the Reviewer’s convenience and all the comments have been addressed:
- According to the Reviewer’s suggestion, we improved Figure 2 by evidencing the different pathophysiological mechanisms, associate with OP driven by inflammaging, post-menopause and use of glucocorticoids. Please, see new Figure 2.
- Thanks for the observation. According to the Reviewer’s comment, we improved section 4 by highlighting the possible benefits and limitations of a Gal-targeted therapy for OP caused by inflammaging compared to the clinical options actually used for general OP. This was done by stressing the results obtained from preclinical evidence regarding the use of Gal inhibitors to reduce inflammation, oxidative stress and autophagy, and from clinical evidence regarding aging, fibrosis and osteosarcoma.
- Thanks for the suggestion. In order to better elucidate the rationale for the selection of Gal-1 and Gal-3 specific subtypes in inflammaging-driven OP, we added a paragraph in section 3.1 reporting:
- the bone-specific expression and activity of Gal-3;
-the importance of intracellular Gal-1 and Gal-3 in BMSCs as progenitors of osteoblasts and regulator of bone formation;
- the central role played in bone differentiation and inflammation by Gal-1 and Gal-3 secreted by BMSCs;
- the main role of Gal-1 and Gal-3 in other pathways (T-cell activation and FVIII) related to bone homeostasis.
- According to the Reviewer’s request, a limitation section has been adding to conclusions, by reporting the limited availability of clinical studies, the presence of contradictory preclinical data and the absence of Gal specific modulators for bone tissue. Furthermore, a description of future directions to facilitate translational research in this area has been added.
Moreover, English has been improved to more clearly express the research and, according to Reviewer 2 indications, some mistakes have been corrected throughout the text.
Reviewer 2 Report
Comments and Suggestions for Authors
In this review, Russo et al. discuss the increasing recognition of the role of inflammaging in the pathogenesis of osteoporosis, as well as the potential use of galectin-targeted agents to modulate inflammaging as a treatment option for osteoporosis. Although direct evidence is still lacking, the review is overall well structured and includes relevant literature. Below are some specific comments that could benefit from further clarification.
Page 4, Line 121:
While bisphosphonate-related jaw osteonecrosis can indeed be complicated by secondary infection, the etiology is thought to be avascular necrosis. Thus, the wording here is slightly inaccurate and may benefit from further clarification.
Page 5, Line 134:
“bybinding” should be corrected to “by binding.”
Page 5, Line 152:
The trial NCT03563235 is testing “Xulin Jiangu granules” rather than “Jiujian Jiangu granules” in comparison to the standard calcitriol arm.
Page 8, Table 1:
The Mercer N et al. (2004) study is an in vitro design, but this information is missing from the corresponding row.
Page 10, Line 309:
NCT02432094 is not a valid trial number. Could the authors verify whether the intended trials are NCT03809052 (for GB1211) and NCT02421094 (for GR-MD-02)?
Author Response
Reviewer 2
The Authors thank the Reviewer for appreciating the Manuscript and for his suggestions in order to improve its quality. All the changes to the Manuscript have been highlighted for the Reviewer’s convenience and all the comments have been addressed:
Page 4, Line 121: Thanks for the suggestion. We rephrased the line, accordingly to the primary and secondary causes of bisphosphonate-related jaw osteonecrosis.
Page 5, Line 134: Corrected.
Page 5, Line 152: Modified, thanks.
Page 8, Table 1: The information has been added to Table 1, as requested.
Page 10, Line 309: Thanks for the correction. We added the NCT numbers suggested. In addition, to provide all the details to reader, we also indicated NCT04607655 for GB1211 compound tested in steatohepatitis, although the clinical study has been classified as “withdrawn”.
Moreover, Figure 2 and text have been improved according to Reviewer 1 suggestions.